# Associations of Dietary Protein Intake and Amino Acid Patterns with the Risk of Diabetic Kidney Disease in Adults with Type 2 Diabetes: A Cross-Sectional Study

**DOI:** 10.3390/nu17132168

**Published:** 2025-06-29

**Authors:** Shih-Ping Lin, Chiao-Ming Chen, Szu-Han Chiu, Po-Jen Hsiao, Kuang-Ting Liu, Sing-Chung Li

**Affiliations:** 1Department of Dietetics, Taoyuan Armed Forces General Hospital, Taoyuan City 32551, Taiwan; ping06072008@gmail.com; 2School of Nutrition and Health Sciences, College of Nutrition, Taipei Medical University, Taipei City 11031, Taiwan; 3Department of Food Science, Nutrition, and Nutraceutical Biotechnology, Shih Chien University, Taipei City 10462, Taiwan; charming@g2.usc.edu.tw; 4Division of Endocrinology, Department of Medicine, Taoyuan Armed Forces General Hospital, Taoyuan City 32551, Taiwan; szu123@aftygh.gov.tw; 5Department of Internal Medicine, Taoyuan Armed Forces General Hospital, Taoyuan City 32551, Taiwan; ndmc6217316@aftygh.gov.tw; 6Division of Nephrology, Department of Medicine, Tri-Service General Hospital, National Defense Medical Center, Taipei City 11031, Taiwan; 7Department of Pathology and Laboratory, Taoyuan Armed Forces General Hospital, Taoyuan City 32551, Taiwan; shaintane@aftygh.gov.tw

**Keywords:** diabetic kidney disease, type 2 diabetes mellitus, ketogenic amino acids, BCAA/AAA ratio, dietary protein, Cox regression, renal protection

## Abstract

**Background/Objectives:** Diabetic kidney disease (DKD) is a major complication of type 2 diabetes mellitus (T2DM), and recent research highlights that amino acid composition—rather than total protein intake alone—may influence DKD risk. This study aimed to evaluate the associations between dietary protein intake, specific amino acid profiles, and the risk of DKD among adults with T2DM. **Methods:** A total of 378 T2DM patients were enrolled in this cross-sectional study. Dietary intake was assessed via a 24 h recall and a validated semi-quantitative food frequency questionnaire. Nutrient analysis was based on the Taiwanese Food Composition Database. Participants were categorized into three protein intake groups: Group 1 (≤0.8 g/kg), Group 2 (0.9–1.2 g/kg), and Group 3 (≥1.3 g/kg). Cox proportional hazards models were used to evaluate the associations of crude protein, branched-chain amino acids to aromatic amino acids (BCAA/AAA) ratio, and ketogenic amino acid intake with DKD risk. Adjustments were made for age, sex, diabetes duration, and blood pressure. **Results:** While crude protein intake showed no significant association with DKD risk, higher intake of ketogenic amino acids (e.g., leucine and lysine) was consistently and significantly associated with reduced DKD risk (adjusted HR range = 0.698–0.716, *p* < 0.01). Our findings highlight the protective potential of ketogenic amino acids such as leucine and lysine, which were significantly associated with lower DKD risk. The BCAA/AAA ratio also showed a downward trend in DKD risk, though not statistically significant. Kaplan–Meier analysis revealed that moderate protein intake (0.9–1.2 g/kg) corresponded to the most favorable DKD-free survival. **Conclusions:** Our findings suggest that, beyond total protein quantity, the intake of ketogenic amino acids may play a protective role in DKD prevention. Moderate protein consumption combined with higher leucine and lysine intake appears beneficial. These results support incorporating amino acid profiling in dietary strategies for DKD risk reduction. Further longitudinal and interventional studies are recommended to validate these associations.

## 1. Introduction

According to the International Diabetes Federation, approximately 537 million adults were living with diabetes worldwide in 2021, and this number is projected to rise to 784 million by 2045 [1]. The increasing global burden of diabetes highlights the urgent need for preventive strategies and healthy dietary practices. In Taiwan, the prevalence of diabetic kidney disease (DKD) among individuals with type 2 diabetes mellitus (T2DM) rose from 13.32% in 2000 to 17.92% in 2014, and the prevalence of end-stage renal disease (ESRD) requiring dialysis reached 1.47% in 2014 [2]. DKD is a leading cause of ESRD globally and is expected to continue rising [3]. It is estimated that approximately 50% of patients with T2DM and one-third of those with type 1 diabetes (T1DM) will develop DKD during their lifetime. As one of the most frequent and burdensome chronic complications of diabetes, DKD places a substantial physical and economic burden on patients, families, and healthcare systems [4].

Diabetic kidney disease (DKD), also known as diabetic nephropathy (DN), is the primary cause of chronic kidney disease (CKD) and the leading contributor to end-stage renal disease (ESRD) in the developed nations. Histopathologically, DKD primarily affects glomerular filtration units and is characterized by basement membrane thickening, mesangial expansion, endothelial cell damage, and podocyte injury. These structural changes contribute to increased urinary albumin-to-creatinine ratio (UACR) and a progressive decline in estimated glomerular filtration rate (eGFR) [3]. Emerging evidence suggests that both the type and quantity of dietary protein may influence renal outcomes. A longitudinal study by Bernier-Jean et al. found that elderly women consuming higher levels of plant-based protein, particularly from fruits, vegetables, and nuts, experienced slower kidney function decline, whereas most animal proteins (except eggs) showed no such benefit [5,6]. In a cross-sectional study by Oosterwijk et al., increased plant protein intake in T2DM patients was associated with a lower prevalence of renal impairment, and substituting 3% of total energy from animal protein with plant protein was associated with a significantly reduced prevalence ratio (PR = 0.20; 95% CI: 0.06–0.63; *p* = 0.01) [7]. However, the 2022 Kidney Disease Outcomes Quality Initiative (K/DOQI) guidelines concluded that there is currently insufficient evidence to recommend specific types of protein, underscoring the need for further research [8].

Regarding protein intake recommendations, the K/DOQI guidelines suggest a daily intake of 0.8 g/kg ideal body weight for non-dialysis patients with diabetes and chronic kidney disease (CKD), based on limited clinical evidence indicating that lower protein intake (<0.8 g/kg/day) may slow kidney function decline in advanced CKD [8]. On the other hand, some recent studies have challenged protein restriction. In the DIALECT cohort study, Oosterwijk et al. reported that among 382 patients with T2DM, a mean protein intake of 1.22 ± 0.33 g/kg/day was not associated with accelerated renal decline. Conversely, patients consuming <1.08 g/kg/day had a significantly higher risk of renal deterioration (HR 1.63; 95% CI: 1.00–2.65) [9]. Heyman et al. later suggested that the discrepancy between older studies and newer findings may be due to differences in concurrent use of renoprotective medications and other dietary factors such as fat intake, which could modify the effect of protein restriction on albuminuria [10].

Additionally, emerging studies indicate that specific amino acids may differentially affect DKD progression. While some amino acids may accelerate disease, others appear to confer renal protection [11]. Amino acids play important roles in metabolic regulation, including glucose homeostasis, and their dysregulation has been linked to T2DM pathogenesis [12]. Elevated plasma levels of branched-chain amino acids (BCAAs) and aromatic amino acids (AAAs) have been associated with increased risk of T2DM [13,14]. Additionally, dietary methionine restriction has been explored for its potential benefits in improving insulin sensitivity and glucose metabolism in diabetic conditions [15]. To date, no clinical nutrition study has comprehensively evaluated amino acid profiles in relation to CKD outcomes. Therefore, the aim of this study was to assess dietary protein and amino acid intake using a semi-quantitative food frequency questionnaire and investigate their associations with DKD in patients with T2DM.

## 2. Materials and Methods

### 2.1. Study Participants and Data Collection

This cross-sectional study recruited 380 participants with T2DM from the dietetic consultation clinic at the Taoyuan Armed Forces General Hospital in Taiwan. Based on their urine test reports, participants were categorized into a diabetes mellitus group (DM) and a DKD according to their UACR. A UACR ≥ 30 mg/g was considered an early indicator of DKD and served as the inclusion criterion for the DN group. A UACR ≥ 300 mg/g indicated more severe kidney damage and also qualified participants for the DN group.

The inclusion criteria required participants to be adults aged 20 years or older who had been diagnosed with T2DM and were capable of completing the dietary assessment questionnaire. The exclusion criteria included individuals with type 1 diabetes, those undergoing dialysis, pregnant women, participants with incomplete dietary or clinical data, and those with other chronic conditions that could independently affect renal function. These conditions included non-diabetic but kidney-impairing chronic diseases such as hypertension (especially if long-term uncontrolled), glomerulonephritis, polycystic kidney disease, lupus nephritis, chronic urinary tract obstruction (e.g., benign prostatic hyperplasia, kidney stones), long-term use of nephrotoxic drugs (e.g., NSAIDs, certain antibiotics, contrast agents), and HIV-associated nephropathy. These criteria were applied consistently throughout the study to ensure sample homogeneity and minimize potential confounding factors.

To investigate the influence of protein intake on DKD, participants were divided into three groups based on their protein consumption: ≤0.8 g/kg, 0.9–1.2 g/kg, and ≥1.3 g/kg. The association between different levels of protein and specific amino acid intake with DKD was assessed. During the study, participants maintained their usual lifestyle and followed physician instructions. At baseline (week 0), participants underwent anthropometric measurements and completed a 24 h dietary recall and FFQ at the nutrition clinic.

During the study, participants maintained their usual lifestyle and followed physician instructions. At baseline (week 0), participants underwent anthropometric measurements and completed a 24 h dietary recall and FFQ at the nutrition clinic. The collected data included demographic characteristics (such as age, sex, diabetes duration, and medication use), clinical biochemistry, and kidney function indicators. Anthropometric measurements included height, weight, BMI, and waist circumference. Clinical and renal markers included blood pressure, fasting glucose, HbA1c, total cholesterol, triglycerides, creatinine, blood urea nitrogen, and microalbuminuria. Both eGFR and UACR were calculated. Dietary assessments included a 24 h recall, dietary habits, and a semi-quantitative FFQ. The above data were statistically analyzed to explore the correlations between protein/amino acid intake and biochemical as well as renal outcomes in individuals with DKD.

A total of 378 participants with T2DM were included in the final analysis after excluding two individuals with incomplete questionnaire data. The cohort included 189 males and 189 females (each accounting for 50%). Based on UACR, 237 participants (62.7%) were categorized as having diabetes without kidney disease, and 141 (37.3%) met the criteria for DKD. Participants were further stratified into three groups according to protein intake levels: Group 1 (≤0.8 g/kg/day, *n* = 160), Group 2 (0.9–1.2 g/kg/day, *n* = 172), and Group 3 (≥1.3 g/kg/day, *n* = 46). All participants were fully informed of the study procedures, objectives, and the content of the administered questionnaires and then signed informed consent. All T2DM participants were approved by the Institutional Review Board of Tri-Service General Hospital, National Defense Medical Center (Approval No. 202405001). The study framework is illustrated in Figure 1.

### 2.2. Assessment of Habitual Dietary Intake Using a Semi-Quantitative Food Frequency Questionnaire

Dietary intake data were collected through one-on-one interviews conducted by registered dietitians. During the interviews, both the conversation and dietary information were recorded simultaneously to ensure accuracy and completeness.

A structured questionnaire, adapted from previous research, was modified and used to assess habitual dietary intake [16]. A semi-quantitative FFQ was developed to reflect commonly consumed protein-rich foods among the Taiwanese population. These included eggs (e.g., chicken eggs, preserved eggs, salted eggs), meats (e.g., poultry and livestock), seafood and its products such as low-fat marine fish (e.g., sea bass, tuna), medium-fat marine fish (e.g., grouper), other fish (e.g., salmon, mackerel), dried fish products (e.g., anchovies, dried whitebait), crustaceans (e.g., shrimp, crab, lobster), mollusks (e.g., clams, oysters), dairy products (e.g., whole, low-fat, and skim milk), soy products (e.g., tofu, dried tofu), and nuts/seeds (e.g., pistachios, almonds, sunflower seeds, pumpkin seeds).

To comprehensively assess protein intake from various sources, the questionnaire also included food groups such as whole grains, vegetables, and fruits. After drafting, the questionnaire underwent expert review by scholars and clinical professionals to evaluate its validity, applicability, and relevance. Pre-tests were also conducted, and feedback was used to refine and revise the questionnaire. Dietary intake was assessed using a semi-quantitative FFQ with 142 food items, reviewed by experts for validity. A pilot test showed good reliability (Cronbach’s alpha = 0.822). Trained staff conducted face-to-face interviews to reduce recall bias.

### 2.3. Nutrient Computation and Data Quality Assurance

After collecting dietary data through structured interviews, nutrient analysis was conducted to estimate the daily intake of energy and macronutrients. The 24 h dietary recall and FFQ data were entered and analyzed using the latest version of the Taiwanese Food Composition Database. Nutrient calculations included total energy (kcal), carbohydrates (g), proteins (g), fats (g), and subcategories such as animal-based and plant-based protein intake.

To enhance the accuracy of portion size estimation, standardized tools such as life-size food models and illustrated nutrition guides were used during the interviews. The nutrient database allowed for the breakdown of intake by food group and macronutrient source, enabling comparisons among the different protein intake groups (≤0.8 g/kg, 0.9–1.2 g/kg, and ≥1.3 g/kg body weight).

Furthermore, to ensure internal consistency and quality control, all data entry and nutrient computation were double-checked by two independent dietitians. The reliability of the nutrient estimates was confirmed by test-retest analysis, and any discrepancies in portion estimation or nutrient coding were resolved through consensus.

### 2.4. Anthropometry and Blood Biochemical Analysis

At baseline (week 0), participants underwent comprehensive clinical assessments at the outpatient clinic of Taoyuan Armed Forces General Hospital. Anthropometric measurements—including body height, weight, and waist circumference—were obtained using a calibrated stadiometer and digital scale. Body mass index (BMI) was computed as weight in kilograms divided by height in meters squared (kg/m^2^).

Blood pressure was measured in a seated position after participants had rested for at least 10 min in a quiet room. Two readings were taken using an automatic sphygmomanometer (FT-500R, Jawon Medical, Kyungsan, Republic of Korea), with a 10 min interval between each, and the average value was recorded.

Venous blood samples were collected after an overnight fast to analyze biochemical markers. Laboratory tests included fasting plasma glucose (FPG), serum creatinine, total cholesterol, triglycerides, BUN, and glycated hemoglobin (HbA1c). Serum biochemical values were determined using an enzymatic colorimetric method via a Cobas c501 analyzer (Roche Diagnostics, Mannheim, Germany). HbA1c was measured using high-performance liquid chromatography (Spotchem SP-4410, Arkray, Kyoto, Japan). Renal function was evaluated through the eGFR, calculated using the modified Modification of Diet in Renal Disease (MDRD) equation: eGFR (mL/min/1.73 m^2^) = 175 × (serum creatinine, mg/dL)^–1.154^ × (age, years)^–0.203^ × 0.742 (if female). Urine samples were collected on the same day to measure albumin and creatinine concentrations. Urinary protein levels were determined using the sulfosalicylic acid precipitation method, and urinary creatinine was measured using the modified Jaffe’s method. UACR was subsequently calculated to assess kidney status.

### 2.5. Statistical Analysis

Descriptive statistics were used to summarize participant characteristics, including sex, anthropometric measurements, urinary microalbumin, urinary protein, eGFR, BUN, serum creatinine, biochemical markers, BMI, blood pressure, and dietary intake variables.

One-way analysis of variance (ANOVA) was conducted to assess differences among the three protein intake groups (≤0.8 g/kg, 0.9–1.2 g/kg, and ≥1.3 g/kg). The analyzed variables included age, sex, anthropometric measures, BMI, blood pressure, biochemical markers, urinary albumin, crude protein intake, and specific amino acid intake. Additionally, comparisons were performed for BCAAs, AAA, the BCAA-to-AAA ratio, and ketogenic amino acids. Differences between groups were tested using one-way ANOVA, we considered the risk of type I error due to multiple comparisons. Where applicable, post hoc comparisons were adjusted using the Bonferroni correction, followed by the Bonferroni method for post hoc comparisons to adjust for multiple testing.

Cox proportional hazards models were used to evaluate the association between dietary factors and the risk of DKD. The event was defined as the occurrence of DKD, and time-to-event was calculated as the duration in months from the date of T2DM diagnosis to the baseline enrollment in the study. Both univariate and multivariate Cox regression models were performed to estimate hazard ratios (HRs) for DKD in relation to total protein intake and specific amino acid variables. Stratified Cox models were also conducted based on the three protein intake groups to explore subgroup-specific effects. Multivariate models included covariates such as age, sex, duration of diabetes, and blood pressure to adjust for potential confounding factors and evaluate the independent effects of dietary exposures on DKD risk. These specific confounders were chosen based on their known influence on DKD risk: age and diabetes duration (Model 1), plus sex (Model 2), and additionally blood pressure (Model 3).

## 3. Results

### 3.1. Comparison of Demographic, Anthropometric, and Biochemical Characteristics Among Different Protein Intake Groups in Patients with Type 2 Diabetes

Table 1 summarizes the demographic, anthropometric, and biochemical characteristics of the 378 participants stratified by protein intake levels. The average age of all participants was 63.4 ± 11.6 years, with no statistically significant differences observed among the three groups (Group 1: ≤0.8 g/kg, Group 2: 0.9–1.2 g/kg, Group 3: ≥1.3 g/kg; *p* = 0.716). The average height of participants was 160.7 ± 9.0 cm, with Group 1 being the tallest (161.7 ± 9.0 cm), followed by Group 2 (160.5 ± 9.1 cm), though no significant difference was noted. Body weight differed significantly across groups, with Group 1 having the highest average weight (74.4 ± 16.5 kg), and the overall mean weight being 68.3 ± 15.6 kg. BMI also varied significantly among groups, with the highest value observed in Group 1 (28.3 ± 5.2).

Regarding renal function markers, the mean eGFR was 82.4 ± 29.4 mL/min/1.73 m^2^. Group 1 had the lowest mean eGFR (78.0 ± 30.7), followed by Group 2 (84.9 ± 29.0) and Group 3 (88.6 ± 23.8), with the difference being statistically significant (*p* = 0.033). Although eGFR generally declines with age, no significant age difference was observed among groups. Serum creatinine levels also differed significantly (*p* = 0.035), with Group 1 showing lower renal function relative to the others. The average UACR was 208.5 ± 738.7 mg/dL, with Group 2 displaying the highest mean UACR (146.3 ± 548.7 mg/dL), although this difference was not statistically significant (*p* = 0.079).

The mean fasting blood glucose level was 132.8 ± 39.9 mg/dL, with Group 1 having the highest value (136.1 ± 48.0 mg/dL), but no significant difference was found among the three groups (*p* = 0.380). However, hemoglobin A1c (HbA1c) levels differed significantly among groups (*p* = 0.037). Microalbumin levels also showed significant variation (*p* = 0.034), with Group 1 having the highest mean (26.4 ± 77.4 mg/dL) compared to Group 2 (10.8 ± 39.6 mg/dL) and Group 3 (9.8 ± 35.0 mg/dL).

No statistically significant differences were observed among the groups for other biochemical variables, including blood pressure, spot urine creatinine, triglycerides, and total cholesterol.

### 3.2. Comparison of Total Protein and Individual Amino Acid Intake Across Protein Intake Groups

Table 2 presents the daily intake of crude protein and specific amino acids across the three protein intake groups, calculated using data from the semi-quantitative FFQ and the Taiwanese Food Composition Database. The overall mean crude protein intake was 61.2 ± 17.5 g/day. Group 1 (≤0.8 g/kg) consumed 49.1 ± 13.1 g/day, Group 2 (0.9–1.2 g/kg) consumed 66.9 ± 13.7 g/day, and Group 3 (≥1.3 g/kg) consumed 82.3 ± 12.3 g/day. All differences between the groups were statistically significant.

Total hydrolyzed amino acids averaged 58.9 ± 16.7 g/day overall, increasing progressively from Group 1 (47.4 ± 12.7) to Group 2 (64.3 ± 13.1) and Group 3 (78.8 ± 11.7). Similar trends were observed for nearly all individual amino acids, including BCAAs (10.3 ± 2.9 g/day overall; Group 1: 8.3 ± 2.2, Group 2: 11.2 ± 2.3, Group 3: 13.8 ± 2.0) and AAAs (5.5 ± 1.5 g/day overall; Group 1: 4.5 ± 1.2, Group 2: 6.0 ± 1.2, Group 3: 7.4 ± 1.1), both of which showed significant differences across groups.

The BCAA/AAA ratio remained consistent across groups, averaging 1.9 ± 0.0. Ketogenic amino acids (expressed as g per 100 g protein) were significantly higher in Group 3 (11.7 ± 1.8), followed by Group 2 (9.5 ± 2.1) and Group 1 (7.0 ± 1.9), with a total mean of 8.7 ± 2.6. All comparisons showed significant differences, suggesting that higher total protein intake was associated with greater intake of both total and specific amino acids.

### 3.3. Association of Protein and Amino Acid Intake with DKD Risk

Table 3 presents the results of the Cox regression analysis assessing the associations between continuous protein-related variables and the risk of DKD. The results show that higher crude protein intake was significantly associated with a reduced DKD risk (β = −0.001, HR = 0.999, 95% CI: 0.998–1.000, *p* = 0.001). Similarly, greater intake of BCAA (β = −0.002, HR = 0.994, 95% CI: 0.990–0.997, *p* = 0.001), AAA (β = −0.012, HR = 0.988, 95% CI: 0.981–0.995, *p* = 0.001), and ketogenic amino acids (β = −0.008, HR = 0.992, 95% CI: 0.988–0.997, *p* = 0.001) were each independently associated with a lower hazard for DKD.

While the effect size of crude protein intake appears modest, its consistent statistical significance emphasizes the potential relevance of overall protein quantity. More notably, the associations with BCAA, AAA, and ketogenic amino acids suggest that specific amino acid profiles may play a stronger protective role. These amino acids may influence DKD risk through pathways related to improved insulin sensitivity, metabolic regulation, or preservation of renal function. Collectively, these findings underscore the importance of not only protein quantity but also amino acid quality in shaping dietary strategies for kidney health in patients with type 2 diabetes.

### 3.4. Kaplan–Meier Curves for DKD-Free Survival Across Different Crude Protein Intake

Kaplan–Meier survival analysis was performed to assess DKD-free survival across protein intake groups. Participants were stratified based on daily protein intake per kilogram of body weight into Group 1 (≤0.8 g/kg), Group 2 (0.9–1.2 g/kg), or Group 3 (≥1.3 g/kg). The time to DKD onset was calculated from the date of diabetes diagnosis to the development of DKD.

Cox proportional hazards models were applied to estimate os (HRs) and generate the survival curves. As shown in Figure 2, no statistically significant differences in DKD-free survival were observed among the three groups. The HRs for crude protein intake were close to 1, suggesting a weak overall association with DKD progression. Using Group 3 (high protein intake) as the reference, Group 2 (moderate protein intake) showed a lower hazard ratio (HR = 0.545) with a borderline *p*-value (*p* = 0.056), indicating a non-significant trend toward reduced DKD risk. Group 1 (low protein intake) also exhibited a slightly lower risk than Group 3, though without statistical significance.

Overall, while moderate protein intake appeared to show a trend toward better DKD-free survival, the Kaplan–Meier analysis did not confirm statistically significant differences among the three protein intake groups. These results suggest that protein quantity alone may not be a strong determinant of DKD risk without consideration of protein quality and amino acid composition.

### 3.5. Adjusted Cox Models Suggest an Inverse Association Between Ketogenic Amino Acid Intake and DKD Risk

Table 4 presents the results of Cox multivariate regression analyses stratified by three levels of protein intake. In the low-protein intake group (Group 1, ≤0.8 g/kg), higher intake of BCAA, AAA, and ketogenic amino acids was significantly associated with reduced DKD risk across the unadjusted and all three adjusted models (Model-1, -2, -3), with BCAA (HR = 0.982, *p* = 0.001) and AAA (HR = 0.969, *p* = 0.002) in Model-2 showing particularly robust protective effects. In the moderate protein group (Group 2, 0.9–1.2 g/kg), BCAA, AAA, and ketogenic amino acids were significantly associated with lower DKD risk in unadjusted and partially adjusted models, but these associations lost statistical significance in Model-3 (adjusted for age, duration of diabetes, gender, and blood pressure), suggesting the influence of covariates. In contrast, in the high-protein group (Group 3, ≥1.3 g/kg), although HRs for all amino acid categories were below 1, none reached statistical significance, indicating that the protective effect of amino acid intake may be diminished at higher protein consumption levels. Overall, these results suggest that within low to moderate protein intake ranges, higher intakes of BCAA, AAA, and ketogenic amino acids are significantly associated with reduced DKD risk, highlighting their potential nephroprotective role in patients with type 2 diabetes.

## 4. Discussion

DKD is a common microvascular complication in T2DM, affecting approximately 40% of patients and serving as a major risk factor for ESRD and diabetes-related mortality [17,18]. Although intensive glycemic and blood pressure control can slow the decline in renal function, current treatments have shown limited efficacy in preventing the progression of DKD to renal failure [19]. In this study, although Group 1 exhibited a higher UACR, the differences among the three groups did not reach statistical significance. However, the eGFR in Group 1 was significantly lower than that of the other two groups. This difference may be attributed to the fact that the eGFR calculation formula includes age as a parameter, and the mean age of participants in Group 1 was significantly higher than that in the other groups. In patients with T2DM, the exact disease duration is often unclear, as hyperglycemia may have been present for several years before a formal diagnosis is made. Some individuals may already have microalbuminuria or overt proteinuria at the time of diagnosis. According to data from the UK Prospective Diabetes Study (UKPDS), after 10 years of T2DM diagnosis, approximately 25% of patients developed microalbuminuria, 5% developed macroalbuminuria, and 0.8% progressed to serum creatinine > 2 mg/dL or ESRD. On average, the annual transition rate from normoalbuminuric to microalbuminuria was 2.0%, from microalbuminuria to macroalbuminuria was 2.8%, and from macroalbuminuria to serum creatinine > 2 mg/dL or ESRD was 2.3% [18].

According to the ANOVA results, Group 1 (with protein intake ≤ 0.8 g/kg) had a significantly lower protein intake compared to the other two groups in this study. Nutritional guidelines for non-dialysis DKD patients recommend a protein intake of approximately 0.8 g/kg/day, which, compared with high protein intake (>1.3 g/kg or >20% of total energy), may help slow eGFR decline and reduce albuminuria [20,21,22,23]. Amino acids are essential nutrients that play critical roles in regulating metabolic balance. In this study, participants with higher total protein intake also exhibited correspondingly higher intakes of individual amino acids. Furthermore, when calculating the proportion of ketogenic amino acids using different normalization methods, the ratio per kilogram of body weight was found to be highest in Group 3, with statistically significant differences observed among the three groups. Based on these findings, further analyses were conducted to evaluate the associations of specific amino acid groups—namely BCAAs, AAAs, and ketogenic amino acids—with the risk of DKD. Studies have shown that dietary amino acid patterns are associated with CKD incidence and may modulate disease progression [24]. BCAAs, which are metabolized in skeletal muscle to BCKAs, may help regulate oxidative stress and renal injury in diabetic mice [25]. Both insufficient and excessive BCAA levels have been associated with adverse outcomes in renal function, suggesting the need for balanced intake [26,27,28]. Therefore, for non-dialysis-dependent CKD patients, in addition to recommended total protein intake, the balance of amino acid patterns should be considered. Appropriate supplementation with BCAAs and histidine and restriction of tyrosine, methionine, and glutamic acid may serve as nutritional strategies to support kidney function [29,30].

Furthermore, a population-based study in Singapore showed a dose-dependent positive association between red meat intake and ESRD risk [31]. In contrast, soy protein intake was associated with improvements in serum creatinine, phosphorus, and triglyceride levels, and reductions in proteinuria [32,33]. Recent studies have indicated that low-protein dietary interventions, particularly those incorporating plant-based sources and supplemented with ketoanalogues, may help delay the progression of CKD and DKD [34,35]. Additionally, soy protein intake has been positively associated with plasma isoflavone levels, supporting its use as a dietary biomarker for plant-based protein consumption [31]. Moreover, our previous study also demonstrated that soybean products provided better protective effects to decrease the risk of diabetic nephropathy [16].

Asghari et al. reported that specific dietary amino acid patterns were significantly associated with the incidence of CKD, with higher intake of plant-derived amino acids (e.g., threonine, serine, arginine) linked to a lower risk (HR = 0.71, 95% CI: 0.51–0.98), while higher intake of animal-derived amino acids (e.g., leucine, isoleucine, tyrosine) may increase CKD risk. Their study also highlighted that amino acid ratios, such as BCAA/AAA, may play a role in renal metabolic regulation [36]. Furthermore, the source of dietary protein also influences ESRD risk. In our study, the BCAA/AAA ratio exhibited a non-significant but protective trend against DKD. Although the ratio did not reach statistical significance in multivariate Cox models (*p* > 0.05), its components—BCAA and AAA—were individually and significantly associated with reduced DKD risk, particularly in the low and moderate protein intake groups. These findings suggest that evaluating the balance between BCAA and AAA intake may provide additional insights into dietary modulation of DKD risk beyond total protein intake. Future prospective or interventional studies are warranted to explore the predictive and therapeutic potential of the BCAA/AAA ratio in diabetic nephropathy.

Our findings align with previous studies indicating that ketogenic amino acids—such as leucine and lysine—may exert protective effects against DKD. Notably, only leucine and lysine are purely ketogenic, while other amino acids like isoleucine and tryptophan are both ketogenic and glucogenic. Liu et al. demonstrated in diabetic animal models that a low-protein diet supplemented with keto acids significantly reduced oxidative stress and slowed DKD progression. Similarly, Huang et al. reported that keto acid supplementation improved muscle atrophy and corrected autophagy abnormalities in type 2 diabetic rats, suggesting dual metabolic benefits for both the kidneys and skeletal muscle [37]. Ketogenic amino acids may exert protective effects in patients with CKD. Ketoacid (KA) supplementation can be transaminated into their corresponding essential amino acids, helping to meet protein requirements without increasing nitrogen load. This approach may improve protein nutritional status and reduce the generation of uremic toxins. In addition, KA supplementation may inhibit inflammatory and apoptotic processes by suppressing NF-κB and MAPK signaling pathways, thereby slowing the progression of CKD [38,39]. Garneata et al. further showed that a very low-protein diet supplemented with ketoanalogues (VLPD-KA) effectively delayed renal function decline in patients with CKD [34], while Chen et al. highlighted anti-inflammatory properties of keto acid supplementation in DKD management [40]. In addition, low-protein diets combined with keto acids (LPD-KA) have demonstrated both metabolic safety and nutritional adequacy in patients with T2DM and CKD [41] and have been associated with reductions in uremic toxins and oxidative stress [42]. Importantly, our study found that greater intake of ketogenic amino acids was consistently and significantly associated with a lower risk of DKD, particularly among individuals with low to moderate total protein intake. These findings support the potential of ketogenic amino acids as nutritional protectants, and further clinical trials are warranted to confirm their therapeutic value within dietary strategies for diabetic populations.

In addition, abnormal metabolism of BCAAs is associated with insulin resistance and may contribute to renal fibrosis and inflammatory responses through modulation of the mTOR signaling pathway. Research also suggests that BCAAs and their metabolites, such as branched-chain keto acids (BCKAs), may have beneficial effects in managing CKD by supporting protein metabolism and reducing nitrogen load [43]. Barba et al. demonstrated that a low aromatic amino acid diet (LA-AAD) significantly reduced the levels of uremic toxins such as p-cresyl sulfate (PCS) and indoxyl sulfate (IS) in mice with CKD, while also improving renal function, reducing inflammation and fibrosis, and preserving nutritional status [44].

On the other hand, Letourneau et al. further explained that AAA, particularly tryptophan, tyrosine, and phenylalanine, are metabolized by gut microbiota into uremic toxins such as IS and p-cresol, which contribute to systemic inflammation and kidney injury in CKD patients [45]. However, from our study, the BCAA/AAA ratio also showed a downward trend in DKD risk, though not statistically significant.

In this study, Kaplan–Meier survival curves were used to evaluate the relationship between different levels of crude protein intake and the risk of developing DKD. Although no statistically significant difference in DKD-free survival was observed among the three groups (*p* > 0.05), participants in the moderate protein intake group (0.9–1.2 g/kg/day) exhibited the most favorable survival trend, with the slowest decline and an HR of 0.545 (*p* = 0.056), suggesting a potentially protective role of adequate protein intake in renal health. This trend is consistent with the findings of Li et al., who reported that glomerular complement C3c deposition in patients with T2DM and biopsy-confirmed DKD was significantly associated with worse renal outcomes. Kaplan–Meier analysis demonstrated a significantly higher risk of renal progression in patients with C3c deposition (*p* < 0.01), and C3c was identified as an independent prognostic factor through Cox regression [46]. Similarly, in a 12-year cohort study, Zhao et al. showed that T2DM patients with microalbuminuria and preserved eGFR (≥60 mL/min) had significantly better survival, as shown by Kaplan–Meier curves, while advanced age, high systolic blood pressure, low BMI, and impaired renal function were linked to increased mortality [47]. These studies underscore the utility of Kaplan–Meier analysis in DKD risk stratification and highlight the clinical value of renal biomarkers and inflammation-related factors. Our findings further contribute to this understanding by suggesting that protein intake levels, particularly moderate intake, may serve as modifiable nutritional indicators for DKD risk, warranting further validation in prospective trials.

One of the strengths of this study is its detailed analysis of dietary amino acid intake in patients with type 2 diabetes, particularly the inclusion of crude protein and specific amino acid groups—such as ketogenic amino acids, BCAAs, and AAAs—into multivariate Cox regression models to examine their association with DKD risk. However, several limitations should be noted. First, the use of a self-developed semi-quantitative FFQ, though reviewed by experts and pilot-tested, may be subject to recall bias, particularly among older adults. Second, the sample size of 378 participants may limit statistical power, especially for subgroup analyses in the high-protein group. Third, nutrient intake estimates were based on the Taiwanese Food Composition Database, which may not fully reflect newer or processed food items. A fourth limitation of this study is the potential inflation of type I error due to multiple testing across dietary variables. Although some post hoc corrections were applied, comprehensive adjustment methods such as FDR were not used, which may warrant caution in interpreting marginally significant findings. Fifth, although we adjusted for major confounders such as age, sex, duration of diabetes, and blood pressure, we did not conduct stratified analyses based on other potential effect modifiers such as BMI or metabolic status due to sample size constraints. We acknowledge this as a limitation and suggest further research with larger cohorts to explore these potential interactions. In addition, the onset of T2DM may precede clinical diagnosis, and detailed data on antidiabetic regimens were not collected; both have been noted as study limitations. Additionally, data on specific treatment regimens for T2DM (e.g., insulin or oral medications) were not collected in detail in this study, which may have introduced residual confounding. Finally, the cross-sectional study design precludes causal inference, and future prospective or randomized controlled trials are warranted to confirm these findings.

## 5. Conclusions

In conclusion, this study provides novel insights into the association between dietary protein quality—particularly specific amino acid profiles—and the risk of DKD in patients with type 2 diabetes mellitus. While crude protein intake alone showed a borderline association with DKD progression, a moderate intake level (0.9–1.2 g/kg/day) demonstrated a favorable survival trend. More importantly, our findings highlight the protective potential of ketogenic amino acids such as leucine and lysine, which were consistently and significantly associated with lower DKD risk, particularly among those with low to moderate protein intake. Although the BCAA/AAA ratio did not reach statistical significance, its components showed individual protective effects, indicating that the balance of amino acid intake may be more relevant than total protein alone. Kaplan–Meier and Cox regression analyses further support the role of amino acid composition in modulating renal outcomes. These results underscore the importance of integrating amino acid profiles, not just protein quantity, into dietary strategies for DKD prevention. Our findings would benefit from validation in a larger sample and in diverse populations. Future longitudinal studies and clinical trials are warranted to validate these associations and evaluate whether targeted amino acid-based nutritional interventions can effectively delay DKD progression and improve metabolic-renal outcomes in diabetic populations.

## Figures and Tables

**Figure 1 nutrients-17-02168-f001:**
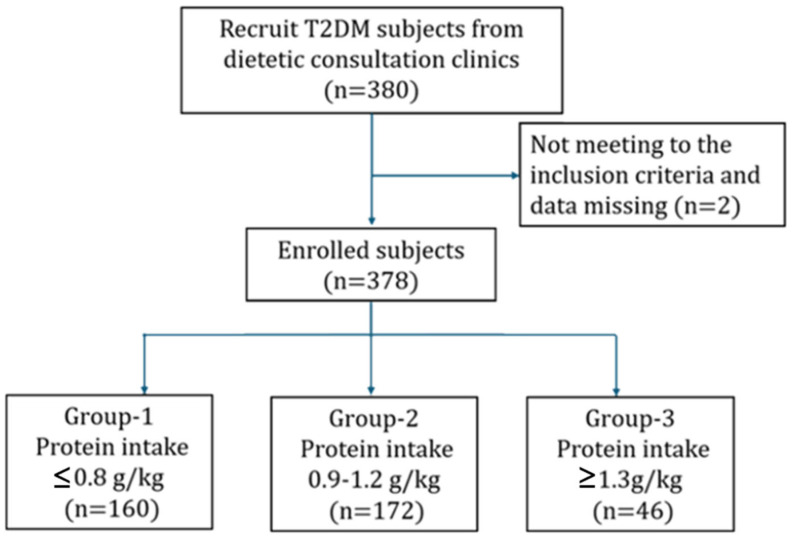
Participant recruitment and assignment process.

**Figure 2 nutrients-17-02168-f002:**
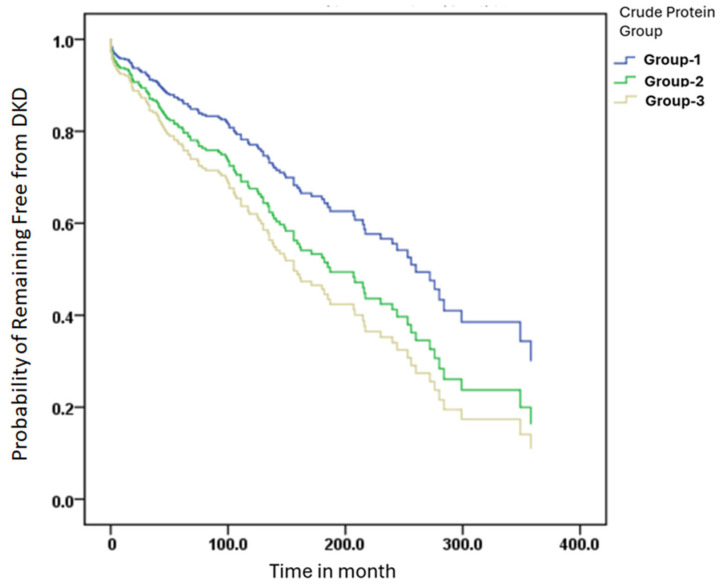
DKD-free survival was analyzed using the Kaplan–Meier method. Protein intake groups were defined by intake per kilogram of body weight: Group 1 (≤0.8 g/kg), Group 2 (0.9–1.2 g/kg), and Group 3 (≥1.3 g/kg). Time to DKD onset was calculated from the date of diabetes diagnosis to study baseline. Cox proportional hazards models were used to estimate hazard ratios (HRs).

**Table 1 nutrients-17-02168-t001:** Baseline demographic, anthropometric, and clinical characteristics of participants, stratified by different protein intake groups according to K/DOQI guidelines.

	All *n* = 378	Group-1	Group-2	Group-3	*p*-Value
		≤0.8 g/kg *n* = 160	0.9–1.2 g/kg *n* = 172	≥1.3 g/kg *n* = 46	
Number (%)	378 (100)	160 (42.8)	172 (45.5)	46 (12.2)	
DM	237 (62.7)	90 (37.9)	114 (48.1)	33 (13.9)	0.068
DKD	141 (37.3)	70 (49.6)	58 (41.1)	13 (9.2)	
Male	189 (50)	23 (12.2)	128 (67.7)	38 (20.1)	0.390
Female	189 (50)	37 (19.5)	127 (67.2)	25 (13.2)	
Age (years)	63.4 ± 11.6	62.9 ± 12.6	63.9 ± 10.8	63.6 ± 11.1	0.716
Body height (cm)	160.7 ± 9.0	161.7 ± 9.0	160.5 ± 9.1	158.3 ± 8.6	0.061
Body weight (kg)	68.3 ± 15.6	74.4 ± 16.5 ^a^	65.9 ± 13.3 ^b^	56.6 ± 10.6 ^c^	<0.001 *
Body mass index (kg/m^2^)	26.2 ± 5.0	28.3 ± 5.2 ^a^	25.3 ± 4.4 ^b^	22.5 ± 2.9 ^c^	<0.001 *
Diastolic blood pressure (mm Hg)	131.2 ± 14.5	132.9 ± 13.8	130.3 ± 14.8	128.9 ± 15.8	0.151
Systolic blood pressure (mm Hg)	77.3 ± 10.5	78.3 ± 10.9	76.4 ± 10.4	77.8 ± 9.3	0.278
Creatinine (mg/dL)	1.0 ± 0.7	1.1 ± 0.8 ^a^	1.0 ± 0.6 ^ab^	0.9 ± 0.8 ^b^	0.035 *
eGFR (mL/min/1.73 m^2^)	82.4 ± 29.4	78.0 ± 30.7 ^a^	84.9 ± 29.0 ^ab^	88.6 ± 23.8 ^b^	0.033 *
UACR (mg/dL)	208.5 ± 738.7	306.9 ± 959.6	146.3 ± 548.7	98.71 ± 336.1	0.079
Microalbumin (mg/dL)	17.3 ± 58.7	26.4 ± 77.4 ^a^	10.8 ± 39.6 ^b^	9.8 ± 35.0 ^b^	0.034 *
Spot urine creatine (mg/dL)	96.7 ± 61.6	103.0 ± 60.7	93.0 ± 62.2	88.9 ± 61.9	0.219
HbA1c (%)	7.4 ± 1.4	7.6 ± 1.5 ^a^	7.2 ± 1.3 ^b^	7.4 ± 1.2 ^ab^	0.037 *
Fasting plasma glucose (mg/dL)	132.8 ± 39.9	136.1 ± 48.0	130.3 ± 33.4	130.5 ± 29.6	0.380
Triglyceride (mg/dL)	143.4 ± 146.7	148.0 ± 88.1	145.5 ± 197.6	119.4 ± 61.4	0.491
Cholesterol (mg/dL)	154.6 ± 38.7	154.0 ± 41.8	152.1 ± 35.1	165.6 ± 39.4	0.108

Values are expressed as mean SD for continuous variables and number (%) for categorical variables. Statistical comparisons across groups were conducted using one-way ANOVA for continuous variables and chi-square tests for categorical variables. Superscript letters (a, b, c) indicate significant differences between groups based on post hoc Bonferroni correction (*p* < 0.05). * DKD according to the American Diabetes Association (ADA) recommends using the term albuminuria for ACR ≥ 30 mg/g.

**Table 2 nutrients-17-02168-t002:** Daily intake of crude protein and specific amino acids among study participants, stratified by different protein intake groups according to K/DOQI guidelines.

	All	Group-1	Group-2	Group-3
		≤0.8 g/kg	0.9–1.2 g/kg	≥1.3 g/kg
Number %	378	160 (15.8)	172 (67.5)	46 (16.7)
Crude protein (g)	61.2 ± 17.5	49.1 ± 13.1	66.9 ± 13.7	82.3 ± 12.3
Total hydrolyzed amino acids (g)	58.9 ± 16.7	47.4 ± 12.7	64.3 ± 13.1	78.8 ± 11.7
Aspartic acid (Asp) (g)	5.4 ± 1.5	4.3 ± 1.1	5.9 ± 1.2	7.2 ± 1.1
Threonine (Thr) (g)	2.4 ± 0.7	1.9 ± 0.5	2.6 ± 0.6	3.6 ± 0.5
Serine (Ser) (g)	2.7 ± 0.7	2.2 ± 0.6	2.9 ± 0.6	3.4 ± 0.6
Glutamic acid (Glu) (g)	10.8 ± 3.1	8.7 ± 2.4	11.8 ± 2.4	14.4 ± 2.3
Proline (Pro) (g)	3.4 ± 1.0	2.7 ± 0.9	3.7 ± 0.8	4.5 ± 0.8
Glycine (Gly) (g)	2.7 ± 0.8	2.2 ± 0.6	3.0 ± 0.6	3.7 ± 0.6
Alanine (Ala) (g)	3.1 ± 0.9	2.5 ± 0.7	3.4 ± 0.7	4.2 ± 0.7
Cystine (Cys) (g)	1.8 ± 0.5	1.5 ± 0.5	1.9 ± 0.4	2.3 ± 0.4
Valine (Val) (g)	2.9 ± 0.8	2.4 ± 0.6	3.2 ± 0.7	3.9 ± 0.6
Methionine (Met) (g)	1.3 ± 0.4	1.1 ± 0.3	1.4 ± 0.3	1.8 ± 0.3
Isoleucine (Ile) (g)	2.6 ± 0.7	2.1 ± 0.6	2.8 ± 0.6	3.5 ± 0.5
Leucine (Leu) (g)	4.8 ± 1.4	3.8 ± 1.0	5.2 ± 1.1	6.4 ± 0.9
Tyrosine (Tyr) (g)	2.2 ± 0.6	1.8 ± 0.5	2.4 ± 0.5	3.0 ± 0.4
Phenyalanine (Phe) (g)	2.7 ± 0.8	2.2 ± 0.6	3.0 ± 0.6	3.6 ± 0.5
Lysine (Lys) (g)	3.9 ± 1.2	3.1 ± 0.9	4.3 ± 1.0	5.3 ± 0.9
Histidine (His) (g)	1.7 ± 0.5	1.4 ± 0.4	1.9 ± 0.4	2.3 ± 0.3
Arginine (Arg) (g)	3.8 ± 1.1	3.1 ± 0.8	4.2 ± 0.9	5.1 ± 0.8
Tryptophan (Trp) (g)	0.6 ± 0.2	0.5 ± 0.1	0.6 ± 0.1	0.7 ± 0.1
Branched-chain amino acids (g)	10.3 ± 2.9	8.3 ± 2.2	11.2 ± 2.3	13.8 ± 2.0
Aromatic amino acids (g)	5.5 ± 1.5	4.5 ± 1.2	6.0 ± 1.2	7.4 ± 1.1
BCAA/AAA	1.9 ± 0.0	1.8 ± 0.0	1.9 ± 0.0	1.9 ± 0.0
Ketogenic amino acids (g)	8.7 ± 2.6	7.0 ± 1.9	9.5 ± 2.1	11.7 ± 1.8

Values are expressed as mean ± SD. One-way ANOVA was used to assess differences among the three groups, followed by Bonferroni post hoc tests. Abbreviations: BCAA = branched-chain amino acids, including leucine, isoleucine, and valine; AAA = aromatic amino acids including phenylalanine, tyrosine, and tryptophan; BCAA/AAA = ratio of BCAA to AAA; ketogenic amino acids, including leucine and lysine.

**Table 3 nutrients-17-02168-t003:** Cox regression analysis of continuous variables involving proteins, BCAA, AAA, and ketogenic amino acids.

Covariate	β	Hazard Ratio	95% CI	*p*-Value
Crude protein (g/kg) ^a^	−0.001	0.999	0.998–1.000	0.001 *
BCAA (g/kg)	−0.002	0.994	0.990–0.997	0.001 *
AAA (g/kg)	−0.012	0.988	0.981–0.995	0.001 *
Ketogenic amino acids (g/kg) ^b^	−0.008	0.992	0.988–0.997	0.001 *

^a^ Crude protein is measured on daily intake divided by body weight (kg). ^b^ Main ketogenic amino acids, including leucine and lysine, are calculated as the total intake in units of daily intake divided by body weight (kg). * *p* < 0.05 was considered statistically significant.

**Table 4 nutrients-17-02168-t004:** Cox multivariate regression analysis based on protein intake of three different intake groups ^a^.

Covariate	β	Hazard Ratio	95% CI	*p*-Value
Group-1 (≤0.8 g/kg)				
BCAA (g/kg)				
Unadjusted	−0.015	0.985	0.974–0.996	0.006 *
Model-1	−0.018	0.982	0.971–0.993	0.001 *
Model-2	−0.018	0.982	0.972–0.993	0.001 *
Model-3	−0.013	0.987	0.976–0.999	0.031 *
AAA (g/kg)				
Unadjusted	−0.027	0.973	0.953–0.993	0.009 *
Model-1	−0.033	0.967	0.948–0.987	0.002 *
Model-2	−0.032	0.969	0.949–0.988	0.002 *
Model-3	−0.023	0.977	0.956–0.999	0.042 *
Ketogenic Amino Acids (g/kg)				
Unadjusted	−0.018	0.982	0.970–0.994	0.004 *
Model-1	−0.021	0.979	0.967–0.991	0.001 *
Model-2	−0.021	0.979	0.967–0.992	0.001 *
Model-3	−0.016	0.984	0.971–0.997	0.018 *
Group-2 (0.9–1.2 g/kg)				
BCAA (g/kg)				
Unadjusted	−0.014	0.986	0.973–1.000	0.043 *
Model-1	−0.018	0.982	0.971–0.993	0.001 *
Model-2	−0.015	0.985	0.972–0.999	0.038 *
Model-3	−0.010	0.990	0.976–1.005	0.198
AAA (g/kg)				
Unadjusted	−0.026	0.974	0.949–0.999	0.044 *
Model-1	−0.033	0.967	0.948–0.987	0.002 *
Model-2	−0.027	0.973	0.948–0.999	0.041 *
Model-3	−0.018	0.982	0.955–1.010	0.211
Ketogenic Amino Acids (g/kg)				
Unadjusted	−0.015	0.985	0.970–0.999	0.042 *
Model-1	−0.017	0.984	0.968–0.999	0.038 *
Model-2	−0.017	0.983	0.968–0.999	0.036 *
Model-3	−0.011	0.989	0.972–1.006	0.191
Group-3 (≥1.3 g/kg)				
BCAA (g/kg)				
Unadjusted	−0.015	0.985	0.962–1.009	0.214
Model-1	−0.042	0.959	0.913–1.007	0.093
Model-2	−0.015	0.985	0.959–1.012	0.270
Model-3	−0.016	0.984	0.953–1.017	0.343
AAA (g/kg)				
Unadjusted	−0.030	0.970	0.927–1.015	0.189
Model-1	−0.041	0.960	0.911–1.011	0.121
Model-2	−0.035	0.966	0.915–1.020	0.210
Model-3	−0.032	0.968	0.910–1.030	0.311
Ketogenic Amino Acids (g/kg)				
Unadjusted	−0.016	0.984	0.959–1.010	0.221
Model-1	−0.021	0.980	0.953–1.007	0.145
Model-2	−0.016	0.984	0.955–1.014	0.281
Model-3	−0.017	0.984	0.951–1.018	0.343

^a^ Multivariate Cox regression analysis identifies independent prognostic factors, and the results indicated that our signature could as independent prognostic factors for overall survival. The Group 1 (≤0.8 g/kg), Group 2 (0.9–1.2 g/kg), and Group 3 (≥1.3 g/kg) were stratified based on total protein intake. Model-1: Variables adjusted for age and duration of diabetes; Model-2: Variables adjusted for age, duration of diabetes, and gender; Model-3: Variables adjusted for age, duration of diabetes, gender, and blood pressure. * *p* < 0.05 was considered statistically significant.

## Data Availability

The original contributions presented in this study are included in the article. Further inquiries can be directed to the corresponding author.

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
