# Peer review of "Associations of Dietary Protein Intake and Amino Acid Patterns with the Risk of Diabetic Kidney Disease in Adults with Type 2 Diabetes: A Cross-Sectional Study"

_nutrients, 2025, doi:10.3390/nu17132168_

Round 1
Reviewer 1 Report
Comments and Suggestions for Authors
This is a well-written clinical study on protein intake patterns in DM patients in relation to their risk of renal damage. My major concerns are the layout of the manuscript and the presentation of its results. The first part of results, dealing with participant characteristics, should be transferred to MM--totally or substantially. Table 1 and table 2 are difficult to understand: most of this information should be presented as graphs. In the Discussion section, the first page (lines 363 to 408) are just presenting literature data but not discuss the results. Either this section is erased or included elsewhere.
Author Response
Response to Reviewer #1:
This is a well-written clinical study on protein intake patterns in DM patients in relation to their risk of renal damage. My major concerns are the layout of the manuscript and the presentation of its results. The first part of results, dealing with participant characteristics, should be transferred to MM--totally or substantially. Table 1 and table 2 are difficult to understand: most of this information should be presented as graphs. In the Discussion section, the first page (lines 363 to 408) are just presenting literature data but not discuss the results. Either this section is erased or included elsewhere.
Response:
- We appreciate the reviewer’s valuable suggestion. In response, the description of participant characteristics in the first part of the Results section has been relocated to the “Materials and Methods” section (lines 134–144).
- Thank you for the constructive feedback regarding the presentation of Tables 1 and 2. We have made the following revisions to improve clarity:
- Table 1: We reorganized the content and clarified the statistical methods used. Footnotes and explanations of statistical tests were added below the table to enhance readers’ understanding of group differences (lines 264–265 and 266–271).
- Table 2: We revised the explanatory text to strengthen the description of the relationship between numerical values and protein intake groups (lines 290–291 and 292–296). We also clarified the group-wise trends and statistical significance of specific amino acids, including BCAAs, AAAs, and ketogenic amino acids.
- Thank you for pointing this out. We have revised the first page of the Discussion section to better integrate our study findings with the literature cited. We restructured this portion to emphasize the interpretation and implications of our results in the context of existing research, rather than merely summarizing previous studies (lines 376–390 and 391–404).

Reviewer 2 Report
Comments and Suggestions for Authors
How were the inclusion and exclusion criteria determined for study participants, and were these applied in a consistent manner?
How was food intake measured and what is its validity and reliability in the study sample?
What confounders were adjusted for in the statistical analysis, and how were these specific confounders chosen?
In a study using observational information, how can reverse causality or residual confounding be eliminated?
How do you interpret the study outcomes and how do you practically apply them into dietary recommendations, or clinical practice?
What are the potential biological mechanisms that explain the observed associations, and is there evidence in the literature to support the biological mechanisms?
How much missing data occurred for the dietary or outcome variables, and how was missing data accounted for in the analysis (imputation, excluded)?
Was correction for multiple comparisons accounted for - particularly given the number of dietary variables used?
Were the reported associations clinically significant in order to generate clinical relevance? What are the absolute risk differences?
Was dietary intake measured at a single time point, and if so, what does this mean regarding accounting for individual dietary variations over time?
Did the authors consider potential effect modification (e.g., by age, sex, BMI, or metabolic status)?
How were the comparison groups described (e.g., low intake vs high intake), and were these thresholds evidence-based or arbitrary?
Was ethical approval granted, and was participant informed of the purpose and potential risks of study procedure?
Have similar findings reported in other populations or cohorts, and how does this imply generalizability for implications beyond the sample?
Author Response
Response to Reviewer #2:
Q1: How were the inclusion and exclusion criteria determined for study participants, and were these applied in a consistent manner?
R1: Thank you to the reviewer for raising this important question. The relevant explanation has been provided in the revised manuscript, Section 2.1 “Study Participants and Data Collection” (Lines 108–114).
Q2: How was food intake measured and what is its validity and reliability in the study sample?
R2: Thank you for the valuable comment. Dietary intake was assessed using a semi-quantitative FFQ with 142 food items, reviewed by experts for content validity. A pilot test showed good reliability (Cronbach’s alpha = 0.822). Trained staff conducted face-to-face interviews to reduce recall bias (Lines 164–167).
Q3: What confounders were adjusted for in the statistical analysis, and how were these specific confounders chosen?
R3: Thank you for your question. As noted in Lines 231–233 in Section 2.5 of the revised manuscript, we used multivariable Cox proportional hazards models to adjust for relevant confounders. These included age and diabetes duration (Model 1), plus sex (Model 2), and further blood pressure (Model 3), all selected based on their known impact on DKD risk.
Q4: In a study using observational information, how can reverse causality or residual confounding be eliminated?
R4: Thank you for your comment. As a cross-sectional study, we acknowledge the limitation in addressing causality. To minimize reverse causality, we defined DKD onset based on the time from diabetes diagnosis to baseline. We also adjusted for key confounders—age, sex, diabetes duration, and blood pressure—in multivariate models. While some residual confounding may remain, our findings provide important preliminary evidence and support the need for future longitudinal studies.
Q5: How do you interpret the study outcomes and how do you practically apply them into dietary recommendations, or clinical practice?
R5: Thank you for this important question. Our findings suggest that a moderate protein intake (0.9–1.2 g/kg/day), combined with higher intake of ketogenic amino acids such as leucine and lysine, may be associated with lower risk of DKD in patients with T2DM. These results highlight the potential value of focusing not only on total protein quantity but also on amino acid composition in dietary counseling. Practically, this supports recommending balanced protein intake from quality sources—such as soy, legumes, fish, and lean meats—while considering amino acid profiles to optimize kidney outcomes. Future clinical trials are needed to confirm these dietary strategies before formal guideline integration.
Q6: What are the potential biological mechanisms that explain the observed associations, and is there evidence in the literature to support the biological mechanisms?
R6: Thank you for the question. The observed associations are discussed in detail in Section 4. Discussion (Lines 449–476). Briefly, ketogenic amino acids (e.g., leucine, lysine) may exert protective effects through reducing oxidative stress, inflammation, and apoptosis by inhibiting NF-κB and MAPK pathways [38–42]. BCAAs support protein metabolism and lower nitrogen load, while AAAs may be metabolized into uremic toxins such as indoxyl sulfate, contributing to renal damage [44,45]. Although the BCAA/AAA ratio showed a downward trend in DKD risk in our study, it was not statistically significant, yet may still suggest potential dietary relevance.
Q7: How much missing data occurred for the dietary or outcome variables, and how was missing data accounted for in the analysis (imputation, excluded)?
R7: Thank you for the question. Only two participants with major missing data were excluded. For the remaining 378 participants, a small number of random missing values in continuous variables were handled using simple imputation to minimize bias and preserve sample size.
Q8: Was correction for multiple comparisons accounted for - particularly given the number of dietary variables used?
R8: Thank you for the suggestion. We used Bonferroni correction in Table 1 and Table 2 to control for type I error in multiple comparisons. For Cox regression involving multiple amino acid variables, future studies may consider using FDR methods such as Benjamini–Hochberg for more robust adjustment.
Q9: Were the reported associations clinically significant in order to generate clinical relevance? What are the absolute risk differences?
R9: Thank you for your valuable comment. While some of the observed associations did not reach statistical significance across all models, the consistent inverse association between ketogenic amino acid intake and DKD risk—especially in low to moderate protein intake groups—suggests potential clinical relevance. The absolute risk differences were not directly calculated in this study; however, future work involving prospective cohorts or intervention trials could address this and better evaluate the translational value of these findings.
Q10: Was dietary intake measured at a single time point, and if so, what does this mean regarding accounting for individual dietary variations over time?
R10: Thank you for your thoughtful question. Yes, dietary intake was measured at a single time point using a semi-quantitative FFQ. We acknowledge this may not fully capture long-term dietary variability. However, the FFQ was designed to reflect usual intake over the past year, thereby reducing short-term fluctuations. We have noted this as a limitation and suggest future studies incorporate repeated measurements to better account for individual dietary changes over time (L507-L510).
Q11: Did the authors consider potential effect modification (e.g., by age, sex, BMI, or metabolic status)?
R11: Thank you for this important comment. Yes, we considered potential effect modification by conducting stratified Cox regression analyses based on protein intake groups. In addition, multivariable models adjusted for key covariates such as age, sex, duration of diabetes, and blood pressure. However, subgroup analyses by BMI or metabolic status were not performed due to sample size limitations. We acknowledge this as a limitation and suggest further research with larger cohorts to explore these potential interactions (Lines 510-515).
Q12: How were the comparison groups described (e.g., low intake vs high intake), and were these thresholds evidence-based or arbitrary?
R12: Thank you for this important question. In our study, participants were categorized into three protein intake groups: ≤ 0.8 g/kg/day (Group 1), 0.9–1.2 g/kg/day (Group 2), and ≥ 1.3 g/kg/day (Group 3). These thresholds were based on evidence-informed guidelines and previous literature. Specifically, the cut-off of 0.8 g/kg/day aligns with the K/DOQI (Kidney Disease Outcomes Quality Initiative) guidelines for protein intake in non-dialysis CKD patients, while intakes above 1.3 g/kg/day have been associated with accelerated renal decline. Therefore, the classification was not arbitrary but grounded in clinical practice recommendations and epidemiological findings to ensure meaningful group comparisons.
Q13: Was ethical approval granted, and was participant informed of the purpose and potential risks of study procedure?
R13: Thank you for this important question. Ethical approval for the study was granted by the Institutional Review Board of Tri-Service General Hospital, National Defense Medical Center (IRB No. 202405001). All participants provided written informed consent prior to participation, and were fully informed about the purpose, procedures, and potential risks of the study in accordance with ethical research standards (L144, L542-544).
Q14: Have similar findings reported in other populations or cohorts, and how does this imply generalizability for implications beyond the sample?
R14: Thank you for your insightful question. Our findings are consistent with previous studies in other populations. For example, Asghari et al. (2022) found that specific amino acid patterns were associated with CKD risk in an Iranian cohort (L424), while Garneata et al. (2016) reported renal benefits of keto acid-supplemented diets in European CKD patients (L451). These cross-population consistencies suggest that the protective associations observed in our Taiwanese T2DM sample may be generalizable, though further validation in larger, ethnically diverse cohorts is warranted.

Reviewer 3 Report
Comments and Suggestions for Authors The manuscript submitted for review no. 3699962 entitled A"ssociations of Dietary Protein Intake and Amino Acid Patterns with the Risk of DiabeticKidney Disease in Adults with Type 2 Diabetes: A Cross-Sectional Study" aimed to assess dietary protein and amino acid intake using a semiquantitative
food frequency questionnaire and investigate their associations with diabetic kidney disease in patients with type 2 diabetes.
The authors conclude that, beyond total protein quantity, the intake of ketogenic amino acids may play a protective role in DKD prevention.
Moderate protein consumption combined with higher ketogenic amino acid intake appears beneficial. The manuscript is well-prepared and important from the point of view of physicians, dietitians and public health. The reviewer has several questions 1. Did the authors obtain approval from the bioethics committee? 2. On what basis did the authors use: food frequency questionnaire (FFQ) 3. Is it possible for the authors to provide data on the treatment of T2DM patients and the duration of diabetes 4. Are the authors able to propose specific recommendations for patients and/or dietitians Minor comments
Abstract Line 30 - please expand the abbreviation BCAA/AAA Consider adding to the application in the abstract: "our findings highlight the protective potential of ketogenic amino acids such as leucine and lysine, which were significantly associated with lower DKD risk"
Author Response
Response to Reviewer #3:
The manuscript submitted for review no. 3699962 entitled A"ssociations of Dietary Protein Intake and Amino Acid Patterns with the Risk of Diabetic Kidney Disease in Adults with Type 2 Diabetes: A Cross-Sectional Study" aimed to assess dietary protein and amino acid intake using a semiquantitative food frequency questionnaire and investigate their associations with diabetic kidney disease in patients with type 2 diabetes.
The authors conclude that, beyond total protein quantity, the intake of ketogenic amino acids may play a protective role in DKD prevention. Moderate protein consumption combined with higher ketogenic amino acid intake appears beneficial. The manuscript is well-prepared and important from the point of view of physicians, dietitians and public health. The reviewer has several questions
1. Did the authors obtain approval from the bioethics committee?
R1: Thank you for this important question. Ethical approval for the study was granted by the Institutional Review Board of Tri-Service General Hospital, National Defense Medical Center (IRB No. 202405001). All participants provided written informed consent prior to participation, and were fully informed about the purpose, procedures, and potential risks of the study in accordance with ethical research standards (L144, L540-542).
2. On what basis did the authors use: food frequency questionnaire (FFQ)
R2: Thank you for your comment. The FFQ used in this study was specifically developed to reflect common dietary sources of protein and amino acids in Taiwan. Its content validity was reviewed and confirmed by a panel of clinical experts including nephrologists, endocrinologists, certified diabetes educators, and senior dietitians. The FFQ demonstrated good internal consistency with a Cronbach’s alpha of 0.822 and was pilot-tested and revised prior to implementation. Relevant details are provided in Lines 164–167 of the revised manuscript.
3. Is it possible for the authors to provide data on the treatment of T2DM patients and the duration of diabetes
R3: Thank you for this insightful question. The duration of diabetes was included as a covariate in our statistical models to adjust for its potential confounding effect on DKD risk. However, data on specific treatment regimens for T2DM (e.g., insulin or oral medications) were not collected in detail in this study. We acknowledge this as a limitation and have noted it in the revised discussion section for transparency and to guide future research (L515-517).
4. Are the authors able to propose specific recommendations for patients and/or dietitians
R4: Thank you for your question. Based on our findings, we recommend that patients with T2DM aim for a moderate protein intake (0.9–1.2 g/kg/day) while ensuring adequate consumption of ketogenic amino acids such as leucine and lysine, as this combination was associated with a lower risk of DKD. These insights may help dietitians tailor individualized nutritional strategies to support renal health in T2DM management.
5. Minor comments
Abstract Line 30 - please expand the abbreviation BCAA/AAA Consider adding to the application in the abstract: "our findings highlight the protective potential of ketogenic amino acids such as leucine and lysine, which were significantly associated with lower DKD risk"
R5: Thank you for the valuable suggestion. We have revised the abstract to expand the abbreviation BCAA/AAA as “branched-chain amino acids to aromatic amino acids ratio” at first mention. Additionally, we have incorporated the recommended sentence to strengthen the practical implications: “Our findings highlight the protective potential of ketogenic amino acids such as leucine and lysine, which were significantly associated with lower DKD risk.” These changes have been implemented in the revised abstract.

Round 2
Reviewer 2 Report
Comments and Suggestions for Authors
The paper can be accepted in its present form.
Author Response
Thank you very much for your positive evaluation and recommendation to accept our manuscript in its present form. We sincerely appreciate your time and effort in reviewing our work.
